# Experts contributions to the development of a non-sugar sweeteners warning label for Brazilian food products

Mariana Ribeiro[1,2,3]*, Priscila de Morais Sato[4], Carlos Felipe Urquizar Rojas[5,6], Carla Galvão Spinillo[6], Laís Amaral Mais[2‡], Camila Aparecida Borges[3,7‡], Ana Paula Bortoletto Martins[1,3]

1 School of Public Health, University of São Paulo (FSP/USP), São Paulo, São Paulo, Brasil, 2 Institute of Consumers Defense (Idec), São Paulo, São Paulo, Brasil, 3 Center for Epidemiological Research in Nutrition and Health, University of São Paulo (Nupens/USP), São Paulo, São Paulo, Brasil, 4 School of Nutrition, Federal University of Bahia (ENUFBA), Salvador, Bahia, Brasil, 5 Design Department of the University of the Joinville Region (UNIVILLE), Joinville, Santa Catarina, Brasil, 6 Design Department of the Federal University of Paraná (UFPR), Curitiba, Paraná, Brasil, 7 Luiz de Queiroz College of Agriculture of University of São Paulo (ESALQ—USP), Piracicaba, São Paulo, Brasil

☉ These authors contributed equally to this work.
‡ LAM and CAB also contributed equally to this work.
* mariana.ribeiro@idec.org.br

## Abstract

The consumption of non-sugar sweeteners (NSS) has been associated with potential health risks, including glucose intolerance, alterations in the intestinal microbiota which lead to metabolic abnormalities including pancreatic endocrine dysfunction, aggravation of kidney disease, increased risk of cancer and adverse cardiovascular outcomes. In Brazil, the presence of NSS in ultra-processed food products (UPFP) is only reported in ingredient lists, which Brazilian consumers struggle to locate and understand. On the other hand, countries such as Argentina and Mexico have implemented a cautionary legend for NSS in packaged foods and beverages. This study aimed to evaluate expert opinion on the design attributes of NSS warning labels intended to inform consumers about the presence of these sweeteners in packaged foods and beverages sold in Brazil. For this purpose, a panel of experts was conducted to discuss, judge and decide on the technical aspects of readability, visibility, attention, perceived healthiness and health risk of the proposed warning labels for NSS in packaged foods and beverages in Brazil. Based on the specific attributes, the experts identified two NSS warning messages as the most suitable: 'Attention: contains non-sugar sweetener — not recommended for children and for weight control' and 'Attention: contains non-sugar sweetener'. According to the experts, these messages should be displayed as front-of-package warnings in black rectangular boxes with white text, beginning with the signal word 'Attention' and positioned on the main panel near the Brazilian front-of-package nutrition labeling. The study provides

**Data availability statement:** All relevant data are within the manuscript and its Supporting information files.

**Funding:** The author(s) received no specific funding for this work.

**Competing interests:** The authors have declared that no competing interests exist.

interdisciplinary evidence and expert insight, to support the development of NSS warning labels in Brazil.

## Introduction

Non-sugar sweeteners (NSS) are a functional class of food additives (FA) defined in Brazil as substances other than sugars that provide a sweet taste to foods and beverages [1]. Originally developed as non-caloric substitutes for sugars [2] they are currently in widespread use, mainly as a component of ultra-processed food products (UPFP), either alone or combined with added sugars [3].

UPFP is a food category defined by the Nova classification [4] and the presence of a NSS is considered a marker to identify UPFP, as they belong to the class of food additives with cosmetic functions. Those types of food additives are used to make the final product palatable or hyper-palatable, and include substances such as flavorings, flavor enhancers, colorings, emulsifiers, emulsifying salts, thickeners, antifoaming, bulking, carbonating, foaming, gelling and glazing agents, besides NSS [3]. Global UPFP consumption has increased over the past decade, and a consolidated body of scientific evidence associates these products with negative health outcomes [5–11]. These findings support the recommendations of the Brazilian Dietary Guidelines, which advise avoiding the consumption of UPFP and prioritizing fresh or minimally processed foods, as well as culinary preparations made from them [12].

In response to growing evidence and following a systematic review and meta-analysis on NSS use [13], the World Health Organization (WHO) published new guidelines in 2023 conditionally recommending against the use of NSS for weight control or to reduce the risk of developing non-communicable diseases (NCD) [14]. These guidelines underscores two gaps in Brazil's labeling policy: the presence of NSS is currently disclosed only in the ingredients list, a section that is often difficult for consumers to locate and interpret [15], and the absence of NSS-specific information in the mandatory front-of-package nutrition labels (FoPNL). Recent evidence associates the consumption of NSS with potential health risks [13], such as glucose intolerance and alterations in the intestinal microbiota, which can lead to metabolic abnormalities (including pancreatic endocrine dysfunction [16–18]. NSS intake has been associated with the worsening of kidney disease [19], as well as an increased risk of cancer [20] and adverse cardiovascular outcomes [21]. Specifically, the consumption of aspartame, sucralose, saccharin and acesulfame potassium are associated with alterations in normal physiological processes involved in the absorption and metabolism of nutrients, contributing to increased abdominal adiposity and body mass index (BMI) in adults and older individuals [22–24].

Given the potential health risks associated with NSS, developing effective strategies to highlight this information on product labels is crucial. Warning labels are an effective strategy to empower consumers by enhancing their understanding of the composition of packaged food products and supporting healthier food choices [25–27], by mitigating the influence of health and nutrition claims [28]. For a warning

to effectively communicate risk, it must include graphic elements that enhance visibility and comprehension [29]. Since the environment of a product package includes branding, marketing elements and nutrition claims [30], warning labels must clearly stand out by using specific design features such as shape, size, typography, and layout to ensure visibility, readability, and comprehension [31]. Color also plays a critical role, with black being the most used color for regulatory warnings in nutrition labeling. The literature further suggests that such messages should begin with a signal word, such as "Attention" or "Caution" [32–35].

Based on a public health and consumer rights approach, Argentina and Mexico have enacted legislation mandating warnings (cautionary legend) for the presence of NSS in packaged foods and beverages [36,37]. These NSS warnings are accompanied by advice against children's consumption and displayed in a black rectangle, with white lettering, making it stand out compared to other packaging elements.

To consider a labeling proposal within the Brazilian FoPNL context, it is necessary to connect with the Chilean experience. Chile was the first country to approve and implement a FoPNL for nutrients in excess (e.g., High in sodium) [38]. The Chilean Food Labeling and Advertising Law represents a significant advancement in this area; however, it lacks a warning to indicate the presence of NSS. Following its implementation, the country experienced widespread UPFP reformulation, characterized by the total or partial replacement of added sugars with NSS, to keep the sweet taste of products. As a result, an increasing number of UPFP became available without the "high in sugar" FoPNL [39,40]. In 2020, Brazil approved a new nutrition labelling regulation, which mandates a FoPNL featuring a magnifying glass icon and the expression "high in" for three nutrients of concern: added sugar, saturated fat and sodium [41,42]. Similar to Chile, the Brazilian FoPNL does not include a warning for NSS and based on the Chilean experience, it is expected that the same reformulation will occur in the Brazilian market.

To uphold the right to information and in light of WHO guidelines and the NSS labeling models in Argentina and Mexico, it is essential to develop a specific NSS labelling strategy for the Brazilian population. This strategy should ensure clear, visible and easy-to-interpret information, enabling consumers to make more informed and potentially healthier food choices. In this context, the present study aimed to evaluate experts' opinions on specific attributes of proposed NSS warning labels designed to inform consumers about the presence of NSS in packaged foods and beverages sold in Brazil.

## Materials and methods

### Study design

A qualitative study was conducted using a panel of experts. This approach is defined as a data collection method for exploratory research, aimed at sharing informed opinions based on current evidence, through discussions and debates among specialists. Its purpose is to support argumentation on emerging or unconsolidated topics, by reaching consensus or identifying key priorities and potential pathways for advancement [43,44]. In this study, the experts panel aimed to discuss, evaluate and decide – at a technical level – on aspects such as legibility, visibility, attention and perception of healthiness and health risk associated with warning labels proposals to inform consumers about the presence of NSS in packaged foods and beverages sold in Brazil.

The warning proposals were developed by the Institute for Consumers' Defense (Idec) and the Laboratory of Information Design Systems at the Federal University of Paraná (LabDSI/UFPR). These proposals were based on the WHO guidelines on NSS use, the warning labels approved by Argentina and Mexico, and discussions within the Southern Common Market – MERCOSUR, as well as the existing warning label literature. Design elements included a signal word, message typographic emphasis and a black box format. The researchers involved in this study were nutritionists and information designers affiliated with public universities or a non-governmental organization (NGO) in Brazil. This study represents the first phase of a broader investigation and aims to support the validation of these models among the Brazilian population.

This study involving human participants was approved by the Ethics Committee of the University of São Paulo (protocol no. 69441423.2.0000.5421). All participants in the expert panel provided written informed consent prior to their participation.

## Sampling

Expert selection considered the following criteria: academic affiliation, regional representation across Brazil and expertise in food and beverage labeling, NSS and/or food advertising. Based on the researchers' professional networks, experts from diverse areas relevant to the study's purpose were nominated. A purposive sample strategy was employed and invitations were sent via email exclusively to Brazilian experts. The exclusive use of the Portuguese language in the proposed labeling models was a consideration, given the potential for variation in interpretation when translated into other languages. The panel included experts from diverse fields such as social communication, information design, law, pharmacy, medicine, nutrition and psychology.

## Material and procedures

The panel of experts was conducted online on October 3, 2023, from 4 pm to 6 pm, via the Google Meet platform. The session was moderated by four researchers involved in the study. An online questionnaire, administered via Google Forms during the meeting, was used to collect individual responses. Experts provided consent for your participation by completing the consent form previously sent. The entire meeting was recorded and transcribed for subsequent content analysis.

The two-hour meeting was divided into three sessions (30 minutes each), in addition to an initial moment (15 minutes) to greet the experts' and firm the agreements, and a final moment (15 minutes) to close the session. The sessions were interconnected, with the outcomes of one influencing the next. To conduct the panel, a mediation guide (S1 File) was developed, covering the following topics: evaluation of the warning message for NSS, evaluation of label readability and evaluation of label visibility in the context of packaging.

Different Google Forms (S2 File) were developed and shared through the meeting chat during each session. Experts submitted their responses anonymously prior to group discussions. Since the session involved the evaluation of visual elements (e.g., label mockups), participants were required to use a laptop or desktop computer. In each session, a Google Form link was shared with the participants and, after completing the questionnaire individually, the experts presented and justified their choices in a group debate. Transition to the next session required agreement on the previous topic, as each step built upon the previous findings. The first session aimed to evaluate the content of the warning message for non-sugar sweeteners, the second to evaluate the readability of the chosen message within a warning model, and the third to evaluate the visibility of the message within a warning model in the context of product mockup packaging.

The first session (evaluation of the NSS warning message) aimed to identify the most appropriate sentence for the NSS warning label, based on the expert's opinions (Fig 1).

Based on the first session's discussion, the second session addressed label readability. Experts assessed the visual presentation of the NSS warning label, with an emphasis on readability and attention capture. Participants were shown different label formats and asked to indicate which version was more readable and more attention-grabbing (Fig 2).

The third and last session (assessment of label visibility in the context of packaging) aimed to analyze NSS warning labels on food and beverage mockups, positioned in different locations. Two previous studies were considered to select the food categories of the mockups: Brazilian Household Budget Survey (*Pesquisa de Orçamentos Familiares* – POF) from 2008–2009 and from 2017–2018 [45], and a Chilean study, which brings the food categories whose use of NSS increased after the implementation of the Food Labeling and Advertising Law [40]. Four product categories were chosen: ready to drink fruit juice, sandwich biscuit, dairy drink and fermented milk. Labels were shown in the following positions:

- **CONTÉM EDULCORANTE**
  (in English: CONTAINS NON-SUGAR SWEETENERS)

- **ATENÇÃO: CONTÉM EDULCORANTE**
  (in English: ATTENTION: CONTAINS NON-SUGAR SWEETENERS)

- **CONTÉM EDULCORANTE - NÃO RECOMENDADO PARA CRIANÇAS**
  (in English: CONTAINS NON-SUGAR SWEETENERS – NOT RECOMMENDED FOR CHILDREN)

- **ATENÇÃO: CONTÉM EDULCORANTE - NÃO RECOMENDADO PARA CRIANÇAS**
  (in English: ATTENTION: CONTAINS NON-SUGAR SWEETENERS – NOT RECOMMENDED FOR CHILDREN)

- **CONTÉM EDULCORANTE - NÃO RECOMENDADO PARA CONTROLE DE PESO**
  (in English: CONTAINS NON-SUGAR SWEETENERS – NOT RECOMMENDED FOR WEIGHT CONTROL)

- **ATENÇÃO: CONTÉM EDULCORANTE - NÃO RECOMENDADO PARA CONTROLE DE PESO**
  (in English: ATTENTION: CONTAINS NON-SUGAR SWEETENERS – NOT RECOMMENDED FOR WEIGHT CONTROL)

- **CONTÉM EDULCORANTE - NÃO RECOMENDADO PARA CRIANÇAS E PARA CONTROLE DE PESO**
  (in English: CONTAINS NON-SUGAR SWEETENERS – NOT RECOMMENDED FOR CHILDREN AND FOR WEIGHT CONTROL)

- **ATENÇÃO: CONTÉM EDULCORANTE - NÃO RECOMENDADO PARA CRIANÇAS E PARA CONTROLE DE PESO**
  (in English: ATTENTION: CONTAINS NON-SUGAR SWEETENERS – NOT RECOMMENDED FOR CHILDREN AND FOR WEIGHT CONTROL)

**Fig 1. Sentences to be evaluated by the experts based on the best message to indicate the presence of NSS in food products.**

**Fig 2. Examples of NSS warning labels to evaluate the readability and attention by the experts.** Note: The texts in Portuguese language are the same as those shown in Fig 1.

(A) main panel near the FoPNL, (B) main panel near the product name, and (C) back panel near the ingredient list and allergen warnings (Fig 3).

In the final form, experts were asked to individually rank each label configuration based on visibility and attention. The session concluded with a final group discussion, informed by the ranking results.

## Data analysis

### Questionnaire analysis

Responses submitted individually via the questionnaires were organized by session and analyzed to identify the results corresponding to each objective. Data related to the individual questionnaires will be indicated below.

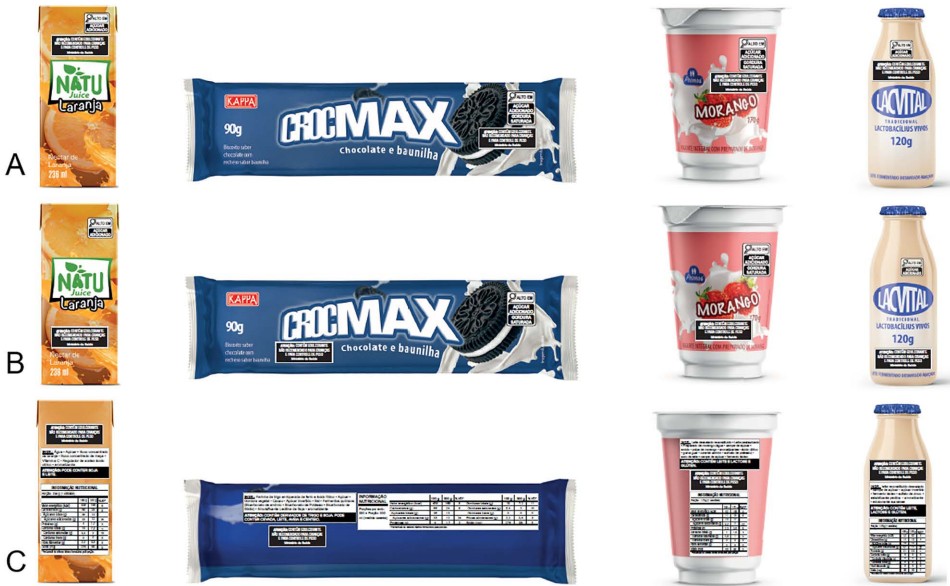

**Fig 3. Food and beverage mockups with different positions of NSS warning label.** (A) main panel and near the FoPNL. (B) main panel and near to sales denomination. (C) back panel and near to list of ingredients and allergen warning.

## Content analysis

The entire panel was recorded and transcribed verbatim. A thematic content analysis approach was used to interpret the experts' discussions [46]. The first author (MR) read the transcript thoroughly, made notes, and identified salient themes. An exploratory analysis was conducted to identify pre-codes (*a priori* and emerging codes), that reflected key aspects of the discussions. These informed the development of the codebook (S3 File). Considering the mediation guide and the exploratory analysis, MR proposed the list of codes, and their definitions, which were discussed with the co-authors until consensus was reached. The finalized codebook included: code; subcode; short name; short and detailed description; inclusion and exclusion criteria; typical and atypical examples; and a "close but no" example to clarify code boundaries. Coding of expert statements was conducted using NVivo 14 software (Lumivero).

## Results

The panel was formed by 10 experts from different knowledge areas, such as: nutrition (n = 4), social communication (n = 1); information design (n = 1), law (n = 1), pharmacy (n = 1), medicine (n = 1) and psychology (n = 1). Participants were from different Brazilian macroregions: Southeast (n = 5), South (n = 2), Northeast (n = 1) and Central-West (n = 1).

Based on the results of the individual questionnaire and the collective discussion, some changes in expert opinion were observed. The predefined questions in the Google Forms focused on technical attributes of design (readability, visibility and attention) and health (perception of healthiness and health risk). In contrast, the collective discussion introduced and prioritized emerging aspects, such as inferences about Brazilian consumers' opinions, design and the expert's technical personal preference.

Regarding the NSS warning label message, "ATTENTION: CONTAINS NON-SUGAR SWEETENER: NOT RECOMMENDED FOR CHILDREN AND FOR WEIGHT CONTROL" ("*ATENÇÃO: CONTÉM EDULCORANTE - NÃO RECOMENDADO PARA CRIANÇAS E PARA CONTROLE DE PESO*", in Portuguese) was selected in the individual questionnaire as the one that makes products seem less healthy (n = 4), makes products seem more harmful to health (n = 6) and attracts

the most attention (n = 6). The second most selected message was "ATTENTION: CONTAINS NON-SUGAR SWEET-ENER" (*ATENÇÃO: CONTÉM EDULCORANTE*, in Portuguese), with the following responses: makes products seem less healthy (n = 3), makes products seem more harmful to health (n = 1) and attracts the most attention (n = 2).

During the collective discussion, the message "ATTENTION: CONTAINS SWEETENER" ("*ATENÇÃO: CONTÉM EDULCORANTE*", in Portuguese) was selected as the most appropriate. The shift in experts' opinion was attributed to several concerns: how Brazilian consumers would interpret a lengthy message, the potential for the message's size to either draw or deter consumer attention in the packaging context, and whether a non-recommendation statement could broaden or narrow the target audience of the warning.

Given the interconnected structure of the sessions, the second session presented two NSS warning label variations, both using the message "ATTENTION: CONTAINS NON-SUGAR SWEETENERS" ("*ATENÇÃO: CONTÉM EDULCOR-ANTE*", in Portuguese). The difference between the proposals was in typographic emphasis: one version had the entire sentence in bold, while the other emphasized in bold only the word "attention" ("*atenção*", in Portuguese), as illustrated in Fig 2. No consensus was reached in this session, so both versions were carried forward to the next one.

In the third session, the same two warning proposals were displayed in the context of packaging (Fig 3). Based on both individual questionnaire (Table 1) and the collective discussion, the position next to the FoPNL was considered the most appropriate.

Considering the content analysis of the collective discussion, the most frequently referenced codes were: attributes of NSS warning labels (n = 26), the attribute of the word "attention" attracting attention (n = 17), use of the word "attention" as a signal (n = 11), and the notion of non-recommendation for NSS consumption (n = 11). The most frequent codes were related to the specific characteristics of the warning label proposals, the design elements and the message of non-recommendation. Conversely, the least frequently referenced codes were: perception of healthiness and/or health risk (n = 7), inference about consumers opinions (n = 9), and expert's personal preference/opinion (n = 9), as shown in Table 2. The limited number of references about perception of healthiness and/or health risk is noteworthy, given that is one of the technical attributes experts were asked to evaluate and considering that seven of the participants were health professionals.

### The NSS warning labels design

The size of the message was a point of discussion, as experts expressed a preference for shorter messages, that are quicker to read and more consistent with other label elements already present on Brazilian packaging (such as information on gluten and lactose). Regarding the visual emphasis, there was a clear preference for presenting the entire message in

**Table 1. Ranking of NSS warning positions on labels.**

| | Message in bold type (n = 8) | | | Word "attention" in bold type (n = 9) | | |
|---|---|---|---|---|---|---|
| **Visibility** | 1° | 2° | 3° | 1° | 2° | 3° |
| Next to the FoPNL | 6 | 2 | 0 | 7 | 2 | 0 |
| After the list of ingredients | 1 | 4 | 3 | 1 | 3 | 5 |
| Next to the sales denomination | 1 | 2 | 5 | 1 | 4 | 4 |
| **Attention** | | | | | | |
| Next to the FoPNL | 6 | 2 | 0 | 7 | 2 | 0 |
| After the list of ingredients | 1 | 2 | 5 | 1 | 2 | 6 |
| Next to the sales denomination | 1 | 4 | 3 | 1 | 5 | 3 |

FoPNL: front-of-package nutrition labeling.

bold, especially given the most appropriate placement for visibility and attention, according to the experts, was near the FoPNL, which is also in bold, or after the list of ingredients.

*"what would be, maybe it would more visible or be more informative, it would be a longer message, on attention and such for children and weight loss, but I keep thinking in terms of visual space on these packages, on these labels, that a very long text will need to use very small print and then it won't draw attention"* (woman, social communication experts).

**Attention (attracting attention and the use of a signal word).** Considering that some of the NSS warning labels adopted the signal word "attention", and that this was a specific attribute to be evaluated by experts, the attention code was subdivided into two subcategories: "the attribute of drawing attention" and "use of the word attention as a signal". For the first subcategory, participants associated the ability to attract attention with multiple elements, such as: the use of specific terms ("attention" and "not recommended"), bold formating, proportionality of the message, and label positioning. As for to the use of the word "attention" as a signal, its placement at the beginning of the message was seen as a strategy that could trigger awareness and convey a sense of alert.

During the second session, when readability and attention of two NSS warning models were analyzed individually, most experts agreed via questionnaire that displaying the entire message in bold improved readability (n = 7) and increased attention (n = 8).

*"(...) the criterion was on getting people's attention, right? Showing that something can be, can present a risk. I think the word 'ATTENTION' makes a difference. I, we have heard this from some people and particularly from me too, if I see the word 'ATTENTION', that already turns on a red light in our head. So that was the first point, having 'ATTENTION' at the front."* (woman, nutrition expert).

**Table 2. Codes, subcodes, description and number of references that emerged through the content analysis of the panel of experts.**

| Code | Subcode | Description | References |
|---|---|---|---|
| Attributes about NSS warning labels | – | Experts' statements regarding the attributes present within the labeling models (message, types of presentation, alert size etc.) proposed to indicate the presence of NSS and on packaged foods and beverages. | 26 |
| Attention | The attribute of attracting attention | Experts' statements regarding the NSS warning labels attract attention when they are seen. | 17 |
| | Use of the word attention as a sign | Experts' statements regarding the use of the term "attention" in the NSS warning labels. | 11 |
| Non-recommendation to consume NSS | – | Experts' statements with arguments not recommending the consumption of NSS for different audiences, contemplating scientific evidence arguments. | 11 |
| Inference about consumers opinions | – | Experts' statements with inferences about the possible opinion and/or interpretation of consumers regarding the labeling models for NSS in packaged foods and beverages. | 9 |
| Expert's personal preference/opinion | – | Experts' statements indicating their personal and/or professional preference/opinion on the NSS warning labels, considering possible mentions or disagreements regarding the questions asked in the panel dynamics. | 9 |
| Perception of healthiness and/or health risk | – | Experts' statements regarding their perception of healthiness and/or health risk in relation to the NSS warning labels. | 7 |

NSS: non-sugar sweetener.

Regarding design attributes, the use of the word "attention" at the beginning of the message and bold formatting were considered differentials that helped to emphasize the message and triggered a sense of alertness among the experts, which they believe could extend to Brazilian consumers. In addition, placing the NSS warning label next to the FoPNL was considered a strategic choice to improve consumer visibility and understanding, even for those unfamiliar with the term NSS.

**The inclusion of a non-recommendation sentence.** Some proposed warning labels included a sentence recommending against NSS consumption by children and/or people attempting to control body weight. Experts noted that this phrasing could either broaden or restrict the perceived target audience. For instance, some consumers might assume that if a product is not suitable for children, it might also be unsuitable for adults, while others might ignore the warning if they are not part of the specified groups. This raised concern among experts, as NSS are not recommended for other groups as well.

*"If I'm not worried about weight control,and I'm not a child, I would consider, right, that this product would be recommended for me, (...) and it's not. So I would also like to know, as a healthy adult, I would like to know that that product is not meant for me"* (man, pharmacy expert).

It is worth highlighting that the statements about the non-recommendation message included healthiness arguments, since the absence of consumption recommendation is related to a potential health risk or benefit.

### Inference about consumers' opinions

Although the panel was designed to gather technical input, some responses reflected assumptions about consumers. Experts believed that consumers are less likely to read long sentences and are more impacted by short messages. The use of audience-specific phrases like "children" or "weight control" generated mixed reactions, because they could both restrict (disregard those who were not mentioned) or expand (understanding that, as it is not adequate for children, it is not suitable for anyone) the target audience. Finally, the main aspect highlighted about this topic was that the specific term for NSS in Portuguese (*edulcorante*) is not understood by the Brazilian population.

*"(...) I stopped at the NSS, I noted 'ATTENTION: CONTAINS NON-SUGAR SWEETENER'. I was very uncertain, but what made me select this option was because the feeling I have is that when we specify the audience, it can exclude me from the audience for whom it is harmful. So, I can think that if this is not suitable for children and those who are controlling their weight, then maybe this is not suitable for me, a person who is not a child and who is not controlling weight (...)"* (woman, psychology expert).

Considering the experts' inference regarding how consumers may interpret labeling terminology, the assumption that Brazilian consumers are unfamiliar with the term "edulcorante", consumers struggle to identify and interpret NSS on food labels, often due to unfamiliar technical terminology and the use of distinct terms [15].

### Expert's personal preference

Some experts noted a discrepancy between their personal preferences and the answers they submitted, as the structured format of the questionnaires did not provide room for open-ended opinion sharing.

*"What I missed when I saw the block of questions was "what is your opinion on this? (...)"* (woman, nutrition expert).

### Perception of healthiness and/or health risk

Regarding those attributes, the experts used elements of the NSS warning labels for discussion. The term "attention" and the use of the non-recommendation phrase were elements indicated as important to interfere in the perception of the

healthiness of products, especially the non-recommendation message that covers two audiences, children and people implicated in controlling body weight. In relation to health risk, only the non-recommendation for children was considered impactful. The use of too many terms on labels was mentioned as a possibility that led to a reductive view of products. Additionally, products without such warnings could be perceived as healthier, which is not always accurate. While the intention of warning labels is to inform, some experts noted that shaping consumer perceptions about health risks could be seen as exceeding that function.

*"The criterion was the question of getting people's attention, right? To show what something can be, to present a risk. I think the word 'ATTENTION' makes a difference. I, we have heard this from some people and particularly from me too, if I see the word 'ATTENTION', that already lights up a red light in our head. So that was the first criterion, having the word 'ATTENTION' at the front. And the other was: It is not recommended for both children and weight control. So, among the options we had, it would be, let's say, a more complete sentence from the point of view of the less healthy, right?"* (woman, nutrition expert).

Although health was an underlying theme throughout the discussions, experts predominantly focused on the design features that shaped or diminished the products perception of healthiness, which aligns with the panel aim. Nonetheless, the central role of healthiness as a guiding element in the rationale for such labeling was not consistently prioritized in the experts' statements. This observation is particularly relevant considering that discussions about health are fundamental from a public health perspective.

The decision to develop a proposal similar to those implemented in Argentina and Mexico reflects an effort to promote uniformity across countries. Specific design elements were identified by the experts as effective features, and a preference for placement in a highly visible position, consistent with established regional practices. While the structured nature of the questionnaire focused on specific attributes, some experts expressed a desire to share broader perspectives. Concerning the messaging, individual questionnaire responses revealed a preference for an extended message format. However, during the collective deliberation, experts favored a more concise warning message that omitted any non-recommendation phrasing. Nonetheless, the final messaging approach selected by the experts in collective diverges from the warning label formats currently enforced in regional countries.

## Discussion

In Brazil, food labeling regulation is coordinated by Anvisa, the national agency responsible for regulating, overseeing and inspecting products and services that pose potential risks to public health. The regulatory process includes the contributions from government, academia, civil society organizations and industry. The most recent review of nutrition labeling regulations, which lasted six years, approved the mandatory FOPNL [41,42]. Despite this, a specific warning for NSS was not adopted with Anvisa stating that NSS labeling would be defined in the MERCOSUR agenda. Currently, Anvisa is leading a new regulatory agenda to revise NSS labeling and use, which is a concrete window of opportunity to discuss this topic. In 2024, the agency published a regulatory discussion paper and held technical meetings with stakeholders, including the food industry, academia and civil society, which highlights the potential for improvement of NSS labeling and proposes the inclusion of a front-of-package warning, particularly addressing contraindications for children [47]. The results of this study will be presented in due course within the ongoing regulatory process, with the aim of informing decisions related to public health and the right to information. It is important to recognize that this proposal may be vulnerable to industry interference, as evidenced during the nutrition labeling process [48]. In addition to the labeling proposal, it is essential that the government commits to implementing public awareness campaigns on the topic, particularly to educate the population about the true meaning of the term "edulcorante".

The use of food additives in products has been increasingly studied in the last decade. A study assessing the list of ingredients of foods and beverages available on the Brazilian market identified that of 9,856 products, 20.6% contain no food additives, 11.6% contained one additive, 19.8% contained between two and three, 23.2% contained between four and five and 24.8% six or more. The study also identified patterns of food additives combinations, with the fifth pattern being exclusively represented by NSS, which was associated with the following product categories: sugar and other non-caloric sweeteners, fruit-flavored drinks and other beverages. Currently, the most frequent food additives category in products sold in Brazil is flavoring [49]. This highlights the importance of improving information about NSS, particularly since such details are currently confined to the ingredients list, which, despite being regulated, pose comprehension barriers for many consumers [15].

Several factors support the call for more transparent labeling of NSS in Brazil. These include the Chilean experience [39,40]; increasing evidence on the potential health risks associated with NSS consumption [13]; the WHO`s conditional recommendation regarding NSS intake [14]; and the absence of any specific NSS warning in the current Brazilian FoPNL system [41,42].

In contrast to regulated information, advertising claims on packaging often receive greater visual emphasis. Evidence shows that visual elements on labels interfere in the perception of products' healthiness [50,51], while FoPNL implementation has been linked to decreased perception of healthiness. To date, however, there are no studies that evaluated Brazilian consumers' perception of NSS warning labels.

One study involving 448 Brazilian consumers found that FoPNL reduced the perceived healthfulness of certain packaged products (e.g., chocolate milk with granola, mate tea with cookies). However, familiar brands and textual nutritional marketing claims also have a significant effect on the perception of healthiness [52]. This demonstrates that while FoPNL helps consumers identify highlighted nutrients, marketing strategies still shape perceived healthfulness.

In Mexico, studies have found that NSS warning labels reduce parents' perceptions of a product's healthfulness more effectively than excess sugar warning alone [53], and the presence of NSS warning influences the healthiness perception of beverages for youth [54]. Such findings support the potential role of NSS-specific warning labels in improving consumer understanding of product nutrition content. It is important to highlight that the presence of NSS in a product characterizes it as an UPFP, a group consistently associated with negative health outcomes [5–11].

While evidence on NSS-specific labeling remains limited, data from Mexico suggest that such labels can reduce the perceived healthiness of UPFPs. Given the likelihood that UPFPs in Brazil may be reformulated by replacing added sugars with NSS following FoPNL implementation, it is crucial to ensure that consumers are provided with clear, accessible information.

The findings from this expert panel represent the initial phase of a broader study designed to assess the effectiveness of NSS warning labels in enabling the identification of NSS in packaged foods and beverages in Brazil. These results will inform the subsequent phase, which involves conducting focus groups with consumers from three distinct regions of the country. Data obtained from the qualitative phases will guide the design of a final quantitative study involving a nationally representative sample of Brazilian consumers.

This study has limitations. Conducting the expert's panel virtually may have hindered the ability to fully capture non-verbal communication, which could indicate unspoken disagreement, especially in instances where a dominant view was expressed by most participants. Although consensus was not a requirement, it was reached in first and third session. In the second session, the absence of consensus was not detrimental, as it resulted in two proposals being forwarded to session three for further analysis. Additionally, despite efforts to diversify the group, it was challenging to recruit experts from different fields across the country with specific expertise on the topic. Nonetheless, purposive sampling enabled the inclusion of a diverse panel of professionals based in various Brazilian cities.

## Conclusion

The study provides interdisciplinary evidence and expert insight, to support the development of NSS warning labels in Brazil. Considering the individual questionnaire and the content analysis of collective discussion, the experts' decision about the most appropriate NSS warning label was: "attention: contains non-sugar sweetener - not recommended to children and for weight control" (Atenção: contém edulcorante – não recomendado para crianças e para controle de peso, in Portuguese) and "attention: contains non-sugar sweetener" (Atenção: contém edulcorante, in Portuguese). Experts recommended that these messages be displayed in bold and positioned on the main panel, near the FoPNL.

## Supporting information

**S1 File. Panel of expert's guide.** Facilitation script used in expert panel sessions.
(PDF)

**S2 File. Panel of expert's form.** Questionnaire items and format used in the Google Form.
(PDF)

**S3 File. Expert panel codebook.** Codebook used for the qualitative content analysis of experts' responses.
(PDF)

## Acknowledgments

To all the experts who voluntarily participated in the panel and shared their knowledge of this study and to the Pan American Health Organization (PAHO).

## Author contributions

**Conceptualization:** Mariana Ribeiro, Carlos Felipe Urquizar Rojas, Carla Galvão Spinillo, Laís Amaral Mais, Camila Aparecida Borges, Ana Paula Bortoletto Martins.

**Data curation:** Mariana Ribeiro.

**Formal analysis:** Mariana Ribeiro, Priscila de Morais Sato, Ana Paula Bortoletto Martins.

**Funding acquisition:** Laís Amaral Mais, Ana Paula Bortoletto Martins.

**Investigation:** Mariana Ribeiro.

**Methodology:** Mariana Ribeiro, Carlos Felipe Urquizar Rojas, Carla Galvão Spinillo, Camila Aparecida Borges, Ana Paula Bortoletto Martins.

**Project administration:** Ana Paula Bortoletto Martins.

**Resources:** Laís Amaral Mais, Ana Paula Bortoletto Martins.

**Software:** Mariana Ribeiro.

**Supervision:** Ana Paula Bortoletto Martins.

**Validation:** Mariana Ribeiro, Priscila de Morais Sato, Carlos Felipe Urquizar Rojas, Ana Paula Bortoletto Martins.

**Visualization:** Mariana Ribeiro.

**Writing – original draft:** Mariana Ribeiro.

**Writing – review & editing:** Mariana Ribeiro, Priscila de Morais Sato, Carlos Felipe Urquizar Rojas, Carla Galvão Spinillo, Laís Amaral Mais, Ana Paula Bortoletto Martins.

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
