## [Decision Letter · Decision Letter 0]

24 Jun 2025

PONE-D-25-27597Experts contributions to the development of a non-sugar sweeteners warning label for Brazilian food productsPLOS ONE

Dear Dr. Ribeiro,

Thank you for submitting your manuscript to PLOS ONE. After careful consideration, we feel that it has merit but does not fully meet PLOS ONE’s publication criteria as it currently stands. Therefore, we invite you to submit a revised version of the manuscript that addresses the points raised during the review process.

**ACADEMIC EDITOR: **

**Required Revisions for Acceptance:**

**1. Abstract** : Clearly highlight the main findings of the study, including the two preferred warning messages and their key design attributes.

**2. Introduction:**

- Improve the transition between the global health context and the Brazilian regulatory landscape by explicitly connecting WHO guidelines to the national policy gap earlier in the text.

- Revise phrasing for conciseness where applicable (e.g., “Recent scientific evidence has associated” → “Recent evidence associates”).

**3. Materials and Methods:**

- Provide more detailed information on how experts were identified and selected (e.g., through professional networks, databases, or institutional affiliations) to ensure reproducibility.

- Discuss how the lack of non-verbal communication in virtual panels may have impacted consensus-building.

- Include examples of the questionnaire (e.g., Google Forms) as supplementary material to enhance transparency.

**4. Results:**

- Support expert assumptions about Brazilian consumers’ understanding of the term “edulcorante” with references to existing literature on label comprehension, if available.

- Expand on why health risk perception was less emphasized by professionals, considering the potential influence of design prioritization.

- Include a brief comparison between Brazil’s proposed warning labels and those implemented in Argentina and Mexico. Address whether the experts’ preferences align with these regional approaches.

**5. Discussion:**

- Discuss regulatory feasibility, including anticipated barriers such as industry resistance or consumer misinterpretation, and explain how the proposed labels may help to overcome them.

- As the study is described as the “first phase,” clearly outline the next steps, especially the planned consumer validation study, including its expected methodology and scope.

- Eliminate redundant content, particularly repetitive references to international experiences (e.g., Chile’s reformulation policies).

**6. References** : Ensure that all references follow a consistent formatting style, as required by PLOS ONE.

We look forward to receiving your revised manuscript.

Kind regards,

Elma Izze Da Silva Magalhães

Academic Editor

PLOS ONE

Reviewers' comments:

Reviewer's Responses to Questions

**Comments to the Author**

1. Is the manuscript technically sound, and do the data support the conclusions?

Reviewer #1: Yes

Reviewer #2: Yes

2. Has the statistical analysis been performed appropriately and rigorously? 

Reviewer #1: No

Reviewer #2: Yes

3. Have the authors made all data underlying the findings in their manuscript fully available?

Reviewer #1: Yes

Reviewer #2: Yes

4. Is the manuscript presented in an intelligible fashion and written in standard English?

Reviewer #1: Yes

Reviewer #2: Yes

5. Review Comments to the Author

Reviewer #1: The review is an attachment below completely

Abstract: The abstract could better highlight the study’s key findings (e.g., the two preferred warning messages and their design attributes).

Introduction: The transition from global/NSS health risks to Brazil’s labeling gap could be smoother. Consider adding a sentence explicitly linking WHO guidelines to Brazil’s regulatory context earlier.

Reviewer #2: Conducting a qualitative study is quite difficult in almost all fields of science. It was carried out systematically and the method section was very well defined. I congratulate you on your unique work, prepared with utmost care.

6. PLOS authors have the option to publish the peer review history of their article (what does this mean? ). If published, this will include your full peer review and any attached files.

**Do you want your identity to be public for this peer review?** For information about this choice, including consent withdrawal, please see our Privacy Policy .

Reviewer #1: **Yes: ** Hasan Basri

Reviewer #2: No

---

## [Author Response · Author response to Decision Letter 1]

12 Aug 2025

We thank you very much for your thorough and thoughtful review and we truly appreciate the attention dedicated to the corrections, and we are also especially the grateful comment regarding the study design. It is very encouraging to receive such positive feedback. The manuscript has been revised in its entirety, and all reviewers’ comments have been addressed point by point in the Response to Reviewers letter.

---

## [Editor Report · Decision Letter 1]

14 Aug 2025

Experts contributions to the development of a non-sugar sweeteners warning label for Brazilian food products

PONE-D-25-27597R1

Dear Dr. Ribeiro,

We’re pleased to inform you that your manuscript has been judged scientifically suitable for publication and will be formally accepted for publication once it meets all outstanding technical requirements.

Kind regards,

Elma Izze Da Silva Magalhães

Academic Editor

PLOS ONE

---

## [Editor Report · Acceptance letter]

PONE-D-25-27597R1

PLOS ONE

Dear Dr. Ribeiro,

I'm pleased to inform you that your manuscript has been deemed suitable for publication in PLOS ONE. Congratulations! Your manuscript is now being handed over to our production team.

Kind regards,

on behalf of

Dr. Elma Izze Da Silva Magalhães

Academic Editor

PLOS ONE